# Interim Estimates of COVID-19 Vaccine Effectiveness in a Mass Vaccination Setting: Data from an Italian Province

**DOI:** 10.3390/vaccines9060628

**Published:** 2021-06-10

**Authors:** Maria Elena Flacco, Graziella Soldato, Cecilia Acuti Martellucci, Roberto Carota, Rossano Di Luzio, Antonio Caponetti, Lamberto Manzoli

**Affiliations:** 1Department of Medical Sciences, University of Ferrara, 44121 Ferrara, Italy; mariaelena.flacco@unife.it (M.E.F.); cecilia.martellucci@unife.it (C.A.M.); 2Local Health Unit of Pescara, 65124 Pescara, Italy; graziella.soldato@ausl.pe.it (G.S.); roberto.carota@ausl.pe.it (R.C.); rossano.diluzio@ausl.pe.it (R.D.L.); antonio.caponetti@ausl.pe.it (A.C.)

**Keywords:** COVID-19, SARS-CoV-2, vaccine effectiveness, vaccine hesitancy, cohort study, Italy

## Abstract

This retrospective cohort study compared the rates of virologically-confirmed SARS-CoV-2 infections, symptomatic or lethal COVID-19 among the residents of the Italian province of Pescara who received one or two doses of COVID-19 vaccines, versus the unvaccinated. The official data of the National Health System were used, and a total of 69,539 vaccinated adults were compared with 175,687 unvaccinated. Among the subjects who received at least one vaccine dose, 85 infections (0.12%), 18 severe and 3 lethal COVID-19 cases were recorded after an average follow-up of 38 days. Among the unvaccinated, the numbers were 6948 (4.00%), 933 (0.53%) and 241 (0.14%), respectively. The serious adverse event reports—yet unconfirmed—were 24 out of 102,394 administered doses. In a Cox model, adjusting for age, gender, and selected comorbidities, the effectiveness of either BNT162b2, ChAdOx1 nCoV-19 or mRNA-1273 vaccines was higher than 95% in preventing infections (mostly due to B.1.1.7 variant), symptomatic or lethal COVID-19. No differences were observed across genders, and among the 691 subjects who received the second dose of vaccine later than the recommended date. Although preliminary, these findings support current immunization policies and may help reducing vaccine hesitancy.

## 1. Introduction

Coronavirus disease-2019 (COVID-19) is spreading worldwide, and has caused over 3,200,000 deaths so far (5 May) [1,2]. Several vaccines have been developed and are being distributed in most countries [3,4,5,6,7]. In Italy, the vaccination campaign against Severe Acute Respiratory Syndrome CoronaVirus-2 (SARS-CoV-2) began on 27 December 2020 [8], and more than 15 million doses have been administered by the end of April 2021 [9]. Although the available vaccines showed a high efficacy in randomized trials (RCTs) [3,4,5,7], data are needed to quantify their effectiveness on the general population. A few interim or preliminary analyses are available, with estimates of vaccination effectiveness ≥70% (single dose) and ≥90% (two doses) in preventing SARS-CoV-2 infection [10,11,12] and COVID-19 disease [13,14]. 

After four months of mass vaccination in Italy, given the urgency for more data to support current public health policies and potentially reduce vaccine hesitancy [15], we performed an interim analysis in order to evaluate COVID-19 vaccines effectiveness in the entire population of an Italian Province.

## 2. Materials and Methods

### 2.1. Study Design, Population and Outcomes

This retrospective cohort study included all the subjects aged ≥18 years who were resident in the province of Pescara, Italy on 1 January 2021. The exposure was COVID-19 vaccination status (one, two, or at least one dose versus none), and the outcomes were SARS-CoV-2 infection (detected through RT-PCR (Reverse transcription polymerase chain reaction, tested through nasopharyngeal swabs by the accredited laboratories of the Local Health Unit of Pescara), virologically-confirmed COVID-19 disease (diagnosed by a specialist physician and requiring hospital admission), and COVID-19-related in-hospital death.

The vaccination campaign started on 2 January 2021, and Pfizer–Biontech BNT162b2, Oxford–AstraZeneca ChAdOx1 nCoV-19 and Moderna mRNA-1273 vaccines were administered gradually to the population [8].

In order to take into account the time required for seroconversion, the start of follow-up varied depending on the administered vaccine. According to the available RCTs [3,4,5], for the evaluation of the effectiveness of the first dose of vaccine, the follow-up started 14 days after the first dose of BNT162b2 [14] and mRNA-1273 [3], and 21 days after the first dose of ChAdOx1 nCoV-19 [5]. Since most of the vaccines that were administered in the first days of January were BNT162b2, the start of the follow-up of the unvaccinated subjects was set at 17 January 2021.

For the evaluation of the effectiveness of the second dose versus none, those receiving only one dose of vaccine were excluded, and the follow-up started 14 days after the second dose for all vaccines [3,4,5]. As the administration of the second dose of BNT162b2 was scheduled 21 days after the first dose, the first day of follow-up for the vaccinated subjects was set 21 days after 17 January, on 6 February 2021.

For all subjects and all evaluations, the end of the follow-up was 21 May 2021 (day of the data extraction). The vaccinated subjects who received the first vaccine dose less than 14 (BNT162b2 and mRNA-1273) or 21 (ChAdOx1 nCoV-19) days before the end of follow-up were excluded from the analysis of the effectiveness of the first dose; the vaccinated subjects who received the second dose less than 14 days before the end of follow-up were excluded from the 2nd dose analysis. All unvaccinated subjects were censored at 21 May 2021.

Only the outcomes that occurred after the start of the follow-up were included, and both vaccinated and unvaccinated subjects who had a positive swab or a COVID-19 disease before the start of the follow-up were thus excluded from all analyses.

In order to account for some of the main potential confounders of the association between vaccination and COVID-19 or death [16], we used the co-pay exemption database (Italian “Esenzioni Ticket” file) and the administrative discharge abstracts of the last ten years (Italian SDO) to extract the following conditions of each resident—diabetes (ICD-9-CM codes in any diagnosis field—250.xx); hypertension (401.xx-405.xx); major cardiovascular or cerebrovascular diseases (410.xx-412.xx; 414.xx-415.xx; 428.xx or 433.xx-436.xx); chronic obstructive pulmonary diseases—COPD (491.xx-493.xx); kidney diseases (580.xx-589.xx); cancer (140.xx-172.xx or 174.xx-208.xx). All data were extracted after encryption from the official vaccination, demographic, hospital and co-pay exemption datasets of the Local Health Unit (LHU) of Pescara.

### 2.2. Public Health Surveillance

The Italian Ministry of Health provided the general guidelines for the National immunization campaign [8]. According to this plan, each LHU is in charge of administering the vaccines and provide all healthcare services to its residents. The surveillance of vaccine uptake, as well as of SARS-CoV-2 testing and COVID-19 cases are part of the National pandemic response plan, and all the information about vaccines, laboratory tests, demographic, anamnestic and clinical residents’ data are routinely entered into LHU official datasets, updated daily, and are sent to the Italian Institute of Health [17]. Vaccination, SARS-CoV-2 testing, and primary or hospital care are free of charge. Regardless of vaccination status, tests are mandatory for all individuals with suggestive symptoms such as fever or acute respiratory illness, for those that have been in contact with infected persons, and for all individuals returning from travel abroad [8].

### 2.3. Data Analysis

First, the differences in the frequency of each recorded outcome, as well as in the other collected variables among vaccinated versus unvaccinated subjects were examined using a chi-squared test and Kruskal–Wallis test for categorical and continuous variables, respectively. For each outcome, Cox proportional hazards analysis was then used to compute the relative hazards of the subjects who received—(a) ≥1 dose; (b) 1 dose only; (c) 2 doses, versus none. All multivariate models were adjusted for age, gender, hypertension, diabetes, major cardio and cerebrovascular events, COPD, kidney diseases, and cancer, all included a priori. Schoenfeld’s test was used to assess the validity of proportional hazards assumption. A two-sided *p*-value of <0.05 was considered significant for all analyses, which were carried out using Stata, version 13.1 (Stata Corp., College Station, TX, USA 2014).

## 3. Results

### 3.1. Characteristics of the Sample

The flowchart of the study participants is shown in Figure 1. From 2 January to 21 May 2021, a total of 69,539 vaccinated and 175,687 unvaccinated adult residents in the Province of Pescara were included in the analyses of vaccine effectiveness. The mean age and the proportion of females of those who received ≥1 vaccine dose were substantially higher than the unvaccinated (67.6 ± 16.7 versus 47.5 ± 18.0 years, and 56.9% vs. 50.3%, respectively).

Of the vaccinated subjects, 52.8% received only one dose within the follow-up; 68.5% received BNT162b2 vaccine; 24.4% ChAdOx1 nCoV-19, and 7.0% mRNA-1273. Among those who received the second dose, 93.8% were vaccinated with BNT162b2, only 0.1% (*n* = 18) with ChAdOx1 nCoV-19, and the remaining 6.1% received mRNA-1273. Among the vaccinated, 70.2% were aged 60 years or more (included in the elderly category), 8.7% were fragile individuals, 7.7% school workers, 6.9% healthcare workers, and 4.1% armed forces or civil operators.

For all outcomes, the average duration of the follow-up of the subjects receiving ≥1 vaccine dose was approximately 38 days, while it ranged from 98 to 100 days for the unvaccinated subjects. During the follow-up, the B.1.1.7 variant accounted for 82.8% of the positive tests (with peaks of 90% during the last two months of follow-up).

### 3.2. Vaccine Effectiveness

Overall, 7033 SARS-CoV-2 infections have been recorded during the follow-up—6948 (2.87%) among the unvaccinated, and 85 (0.12%) among those who received at least one dose (*p* < 0.001; Table 1). The number of COVID-19 cases was 951; of these, 933 were registered in the unvaccinated group (0.53%), and 18 among vaccinated subjects (0.03%; *p* < 0.001). A total of 244 SARS-CoV-2-positive persons died—241 (0.14%) were not immunized (seven of whom younger than 50 years); three received at least one dose of vaccine (an 84-year old male, and two females, aged 89 and 96 years).

Adjusting for age, gender, and selected comorbidities, vaccination effectiveness was very high for any outcome (Table 1). Compared with the unvaccinated, the hazards of SARS-CoV-2 infection were 98% lower among those who received two doses of BNT162b2 (adjusted Hazard Ratio (HR)—0.02; 95% Confidence Interval (CI)—0.01–0.04); 100% lower after two doses of mRNA-1273 (*n* = 0 infections); and 95% lower for those who received one or two doses of any vaccine (HR: 0.05; 95% CI: 0.04–0.06). The effectiveness after a single dose varied by vaccine, ranging from 55% (BNT162b2) to 95% (ChAdOx1 nCoV-19-Table 2).

Two doses of BNT162b2 or mRNA-1273 were able to reduce by 99% and 100% the hazards of COVID-19 disease (HRs—0.01 and 0.00, respectively). In the sample restricted to those who received only one vaccine dose, the COVID-19 cases were 18 (0.11%), 0 and 0 among those vaccinated with BNT162b2, ChAdOx1 nCoV-19 and mRNA-1273, respectively (vs. 933 among the unvaccinated; 0.53%). Vaccine effectiveness to prevent death was very similar—98% to 100% after two doses of either BNT162b2 or mRNA-1273, with no deaths among those who received a single dose of ChAdOx1 nCoV-19 or mRNA-1273, and two deaths among those who received only one dose of BNT162b2.

While the average follow-up of those receiving a single dose of ChAdOx1 nCoV-19 was 30.4 days, the mean follow-up duration of the subjects who received only one dose of m-RNA vaccines was too short (13.0 days for BNT162b2; 17.4 days for mRNA-1273) to permit a meaningful evaluation of the effectiveness of these vaccines against outcomes, such as COVID-19 or death, that may require 3–20 days to occur after the contagion [18].

All analyses were repeated separately by gender (Table 3)—after two doses, vaccine effectiveness was similar for males (99% against the infection; 100% against COVID-19 or death) and females (97% against the infection, 99% against COVID-19, 96% against death, with overlapping 95% confidence intervals for both outcomes). After one dose, the results were also similar across genders, for all the outcomes.

As an additional secondary analysis, we stratified the subjects who received two doses of vaccine according to the number of days between the first and second dose—those who received the second dose within the time interval recommended by the producers (21–28 days for BNT162b2; 28 days for mRNA-1273 [19]) were compared to those who received the second dose later than the recommended schedule. As shown in Table 4, 691 out of the 32,855 (2.1%) subjects who received two doses of vaccine were immunized with a delayed schedule. The vast majority of them (91.3%) received the second dose within 35 days from the first. Overall, we observed zero infections, diseases and deaths among the 691 subjects with delayed administration of the second dose, versus 17 infections, 1 disease and 1 death among the 32,164 subjects who received the second dose within the recommended time. None of the three outcomes was significantly different between the two groups of vaccinated subjects.

For a total of 102,394 administered doses, the overall number of serious adverse event reports that were received during the follow-up was 24 (with zero deaths; 2.3 × 10,000). The regional network is currently evaluating these reports in order to verify how many of them could have been caused by the vaccination.

## 4. Discussion

In the general population of an Italian province, where 28.4% of the citizens received at least one dose of mRNA (BNT162b2 or mRNA-1273) or adenoviral-vectored (ChAdOx1 nCoV-19) COVID-19 vaccines, and were followed for an average of 38 days after seroconversion, the proportion of virologically confirmed SARS-CoV-2 B.1.1.7 variant infection, COVID-19 symptomatic disease, and COVID-19-related death were drastically lower among vaccinated versus unvaccinated individuals.

Overall, adjusting for age, gender, selected comorbidities, and follow-up duration, the risk of infection decreased by 98% after two doses of vaccine, and by 55% to 95% after one dose, depending on the vaccine type. Vaccine effectiveness was even higher against COVID-19 severe or lethal diseases, which were reduced by 99% and 98%, respectively, after two doses of BNT162b2 or mRNA-1273, or one dose of ChAdOx1 nCoV-19.

These findings are consistent with those of recent observational studies on vaccine protection against severe COVID-19, in the context of mass vaccination campaigns, that reported an effectiveness ranging from 57% to >90% after one dose, and ≥90% after two doses [13,14,20,21]. Our results also expand and confirm previous findings on vaccine effectiveness against SARS-Cov-2 infection—the data available so far documented a risk reduction of 70–84% after one dose, and 85–95% after two doses [10,11,12,14,20]. These estimates were based upon mRNA vaccines, largely employed among healthcare workers and in the early stages of mass vaccinations. Our results confirmed these findings for BNT162b2 and mRNA-1273 vaccines, and, although based on a single administration, suggest a comparably high effectiveness for adenoviral-vectored ChAdOx1 nCoV-19 vaccine. Unfortunately, the design of the study and the scarcity of outcomes among the vaccinated did not permit a direct comparison between the administered vaccines, all of which showed a very high effectiveness, with overlapping 95% confidence intervals most of the time. Also, the average follow-up duration of those who received only one dose of m-RNA vaccines was too short (13.0 days for BNT162b2; 17.4 days for mRNA-1273) to permit a meaningful evaluation of the effectiveness of these vaccines against outcomes that may require 3–20 days to occur after the contagion [18].

Although higher than what is reported by preliminary CDC data (0.5 × 10,000) [22], the rate of serious adverse event reports in our population was very low (2.3 × 10,000), confirming the satisfactory safety profile reported in randomized clinical trials [3,4,5], and the few available interim reports following market authorization, which showed a comparable risk for either mRNA and adenoviral-vectored vaccines [23,24,25].

No significant differences were observed in vaccine effectiveness across genders at any dose and for any outcome. This finding is in line with previous trials and real-world studies on healthcare workers [3,4,14], and may help, at least in part, to shorten the gap in vaccine hesitancy between males and females, the latter having shown higher reluctancy in several surveys [14,26,27,28,29]. Beyond effectiveness, however, the potential reasons behind female hesitancy could be the lower COVID-19 lethality [2,16], and especially the substantially higher rates of adverse events among vaccinated women [23,30]. In this study, the number of serious adverse event reports was too scarce to allow a meaningful analysis of the gender differences. Notably, however, even though the rate of adverse events was substantially higher among females, it was still very low for both genders, which should be clearly communicated by mass vaccination information campaigns. Finally, the lower propension to immunize females did not seem to translate into lower vaccination rates in the few studies available to date [11].

In the current scenario of limited vaccine availability, some EU countries, including Italy, have decided to delay the administration of the second dose of BNT162b2 vaccine (from three to six weeks after the first dose), in order to use all available doses to expand the proportion of population who received at least one dose of vaccine [31,32]. While a longer gap between doses has shown an efficacy comparable to the recommended schedule for the ChAdOx1 nCoV-19 vaccine [5,33], real-world comparisons between different dosing intervals of BNT162b2 and mRNA-1273 vaccines are awaited [34]. In our sample, no differences in the effectiveness of BNT162b2 or mRNA-1273 vaccines emerged among those who received a delayed second dose (*n* = 691), as compared to people vaccinated according the standard regimen [19]. These findings are in line with a preliminary, UK-based pre-print analysis on 172 elderly subjects immunized with the BNT162b2 vaccine [35], in which the antibody response was 3.5-fold higher with a delayed interval vaccination, as compared to the standard, 3-week schedule. Although based upon limited data, thus requiring confirmation, these findings support the current Italian policy on a delayed second dose administration and might be crucial to maximize vaccination benefits in the contest of limited supplies.

The current pandemic scenario is further complicated by the presence of new SARS-CoV-2 variants. Concerns were raised about the efficacy of currently available vaccines, and the durability of the immune response, against the three currently prevalent variants [34]. By now, several laboratory assessments have shown a largely preserved in vitro antibody-neutralizing activity against B.1.1.7 (UK) and B.1.351 (South African) variants, and a lower efficacy against the P.1 (Brazilian) variant [36,37,38], while the few real-life data so far available suggest a preserved effectiveness, both for BNT162b2 and ChAdOx1 nCoV-19 vaccines, against B.1.1.7 and B.1.351 [11,13,14]. We did not specifically investigate vaccine effectiveness across circulating variants. However, given the dominance of the B.1.1.7 variant during the study period, the present findings confirm those of the studies published so far [11,13,14], showing a high effectiveness of the current vaccines against this largely diffused variant.

Taken together, these findings suggest that COVID-19 vaccine benefits largely outweigh their risks. Our results are in line with the available data; however, the duration of follow-up is limited, and validated data on vaccine-linked serious adverse events are still being awaited. Also, although our analyses relied upon a dataset including vaccination and clinical data extracted from routinely collected electronic health records, observational studies are prone to recall or misclassification bias [21], and biases could be introduced as well if testing policies were affected by vaccination status, or restrictive measures were applied differently to vaccinated and unvaccinated subjects. Although misclassification bias is unlikely for COVID-19 cases and deaths, it is possible that some SARS-CoV-2-infected individuals, who reported being asymptomatic at the time of interview, might have instead been presymptomatic. However, as reported by Haas and Colleagues, this type of misclassification was probably uncommon and would be unlikely to substantially influence the vaccine effectiveness estimate against asymptomatic infection [20]. With regard to testing and restriction policies, in Italy, in contrast to other countries [20], vaccinated and unvaccinated subjects were subject to the same testing requirements and restriction policies, thus it is unlikely that the estimates are substantially affected by test frequency or lockdown measures [39]. In any case, the present results are inevitably preliminary and require confirmation, like those of most previous studies [10,11,12,13,14]. However, interim analyses from real-life mass vaccination campaigns are crucial, and were urgently needed, to reinforce current policies on widespread immunization and hopefully reduce vaccine hesitancy [15].

With regard to the latter concept, previous research suggested that some of the key drivers of vaccination uptake are the perceived efficacy of a vaccine, and concerns over serious adverse events [40]. This was particularly true during the COVID-19 immunization campaign, as the fast pace of vaccine development—exceptional but unprecedented–contributed to rise doubts upon vaccination safety and effectiveness [41,42]. However, some large surveys from the UK suggested that the chances of vaccination increased, even among the more reluctant subgroups, once it was acknowledged that the immunization significantly reduced the risk of severe disease [40,43]. Similarly, the probability of being vaccinated was substantially increased after clear information upon the safety profile of a vaccine was provided [43]. In this scenario, our study, based upon real-world data, may have important practical implications for public health policy, as it may help reassure the general population on both the effectiveness and the safety of the most widely used COVID-19 vaccines.

## Figures and Tables

**Figure 1 vaccines-09-00628-f001:**
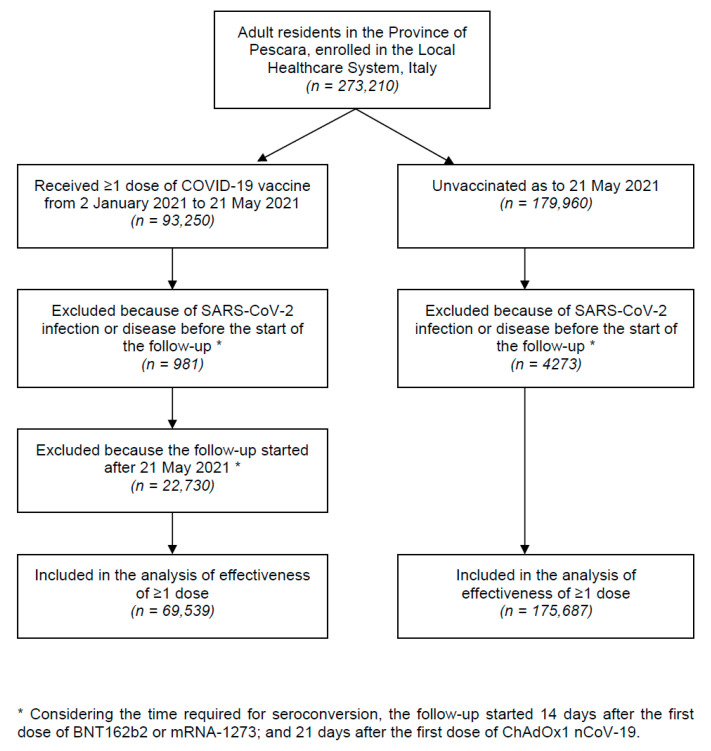
Flowchart of study participants.

**Table 1 vaccines-09-00628-t001:** Main characteristics and outcomes of the sample, overall and by vaccine status.

	Overall Sample	≥1 Vaccine Dose	Unvaccinated	
Variables	(*n* = 245,226)	(*n* = 69,539)	(*n* = 175,687)	*p* *
Mean age in years (SD)	53.2 (19.8)	67.6 (16.7)	47.5 (18.0)	<0.001
Male gender, %	47.9	43.1	49.7	<0.001
Priority category, % ^A^				
- Elderly	35.8	70.2	22.2	<0.001
- Fragile individuals	2.5	8.7	--	--
- School workers	2.2	7.7	--	--
- Healthcare workers	1.9	6.9	--	--
- Armed forces and civil operators	1.1	4.1	--	--
- Others	0.7	2.4	--	--
- Unknown	55.7	0.0	77.8	<0.001
Risk factors and comorbidities ^B^				
- Hypertension	13.9	30.4	7.4	<0.001
- Diabetes	5.8	12.6	3.1	<0.001
- Major cardio/cerebrovascular diseases	7.1	15.2	3.8	<0.001
- COPD	3.0	5.1	2.1	<0.001
- Kidney diseases	1.9	3.5	1.3	<0.001
- Cancer	5.0	10.3	3.0	<0.001
N. of vaccine doses, % ^C^				
- One or two (*n* = 69,539)	28.4	100.0	--	--
- One only (*n* = 36,684)	15.0	52.8	--	--
- Two (*n* = 32,855)	13.4	47.2	--	--
Vaccine type-1st dose, % ^C^				
- BNT162b2 (*n* = 47,654)	19.4	68.5	--	--
- ChAdOx1 nCoV-19 (*n* = 16,997)	6.9	24.4	--	--
- mRNA-1273 (*n* = 4888)	2.0	7.0	--	--
- None (*n* = 175,687)	71.6	--	100.0	--
Vaccine type-2nd dose ^D^				
- BNT162b2 (*n* = 30,817)	12.6	93.8	--	--
- ChAdOx1 nCoV-19 (*n* = 18)	0.0	0.1	--	--
- mRNA-1273 (*n* = 2020)	0.8	6.1	--	--
- None (*n* = 212,371)	86.6	--	100.0	--
Average days between the start of the follow-up and SARS-CoV infection, (SD) ^E^	98 (43)	38 (29)	122 (17)	<0.001
Average days between the start of the follow-up and COVID-19, (SD) ^F^	100 (42)	38 (29)	125 (6)	<0.001
Average days between the start of the follow-up and death, (SD) ^G^	100 (42)	38 (29)	125 (3)	<0.001
SARS-CoV-2 positive swab, % (*n*) ^D^	2.87 (7033)	0.12 (85)	4.00 (6948)	<0.001
Covid-19 disease, % (*n*) ^D^	0.39 (951)	0.03 (18)	0.53 (933)	<0.001
Death, % (*n*) ^D^	0.10 (244)	0.00 (3)	0.14 (241)	<0.001

* Chi-squared test for categorical variables, Kruskal–Wallis test for continuous ones. ^A^ The priority category, besides age, was not available for non-vaccinated subjects. Elderly—subjects aged ≥60 years; Fragile individuals—elderly in long-term care institutions, persons with a list of disabilities and chronic diseases defined by the Italian Government (the complete list is available at https://www.trovanorme.salute.gov.it/norme/renderPdf.spring?seriegu=SG&datagu=24/03/2021&redaz=21A01802&artp=1&art=1&subart=1&subart1=10&vers=1&prog=002 accessed on 9 June 2021; Healthcare workers—physicians and healthcare professionals; students of medicine and healthcare professions; long-term care facilities and clinical lab staff; School workers—all teachers from kindergarten to university; Armed forces and civil operators—policemen, firemen, military, civil protection operators; Others—people cohabiting with fragile individuals or SARS-CoV-2-infected subjects. ^B^ Subjects with the selected comorbidities in the Regional co-pay exemption database (Italian “Esenzioni Ticket” file) or an hospital admission in the last ten years (from the Italian SDO database of administrative discharge abstracts) with the following ICD-9-CM codes in any diagnosis field—250.xx (diabetes); 401.xx-405.xx (hypertension); 410.xx-412.xx or 414.xx-415.xx or 428.xx or 433.xx-436.xx (major cardiovascular or cerebrovascular diseases); 491.xx-493.xx (chronic obstructive pulmonary disease—COPD); 580.xx-589.xx (kidney diseases); 140.xx-172.xx or 174.xx-208.xx (cancers). ^C^ Administered at least 14 (BNT162b2 and mRNA-1273) or 21 (ChAdOx1 nCoV-19) days before the end of follow-up (21 May 2021). ^D^ After 14 (BNT162b2 and mRNA-1273) or 21 (ChAdOx1 nCoV-19) days from the administration of the first dose for the vaccinated subjects, and after 16 January 2021 for the unvaccinated subjects. ^E^ From the start of the follow-up to the first SARS-CoV-2-positive swab or the end of follow-up (21 May 2021). ^F^ From the start of the follow-up to the date of diagnosis of COVID-19 (confirmed with at least one SARS-CoV-2-positive swab), or the end of follow-up (21 May 2021). ^G^ From the start of the follow-up to in-hospital death (only subjects with at least one SARS-CoV-2 positive swab), or the end of follow-up (21 May 2021).

**Table 2 vaccines-09-00628-t002:** Multivariate analysis of the effectiveness of COVID-19 vaccines.

Cox Model ***	SARS-CoV-2	COVID-19	Death
	HR (95% CI)	HR (95% CI)	HR (95% CI)
≥1 Vaccine dose vs. none ^A^			
- None (*n* = 175,687)	1 (Ref. cat.)	1 (Ref. cat.)	1 (Ref. cat.)
- All vaccines (*n* = 69,539)	0.05 (0.04–0.06)	0.04 (0.02–0.06)	0.03 (0.01–0.08)
- BNT162b2 (*n* = 47,654)	0.05 (0.04–0.07)	0.05 (0.03–0.08)	0.03 (0.01–0.09)
- ChAdOx1 nCoV-19 (*n* = 16,997)	0.05 (0.03–0.08)	0.00 (NE)	0.00 (NE)
- mRNA-1273 (*n* = 4888)	0.02 (0.01–0.07)	0.00 (NE)	0.00 (NE)
Only one vaccine dose vs. none ^B^			
- None (*n* = 175,687)	1 (Ref. cat.)	1 (Ref. cat.)	1 (Ref. cat.)
- All vaccines (*n* = 36,684)	0.16 (0.13–0.20)	0.31 (0.19–0.49)	0.27 (0.07–1.10)
- BNT162b2 (*n* = 16,837)	0.45 (0.34–0.60)	NE ^ψ^	NE ^ψ^
- ChAdOx1 nCoV-19 (*n* = 16,979)	0.05 (0.03–0.08)	0.00 (NE)	0.00 (NE)
- mRNA-1273 (*n* = 2868)	0.07 (0.02–0.26)	NE ^ψ^	NE ^ψ^
Two vaccine doses only vs. none ^C^			
- None (*n* = 174,023)	1 (Ref. cat.)	1 (Ref. cat.)	1 (Ref. cat.)
- All vaccines (*n* = 32,855)	0.02 (0.01–0.03)	0.01 (0.00–0.04)	0.02 (0.00–0.12)
- BNT162b2 (*n* = 30,817)	0.02 (0.01–0.04)	0.01 (0.00–0.04)	0.02 (0.00–0.13)
- ChAdOx1 nCoV-19 (*n* = 18)	--	--	--
- mRNA-1273 (*n* = 2020)	0.00 (NE)	0.00 (NE)	0.00 (NE)

HR = Hazard Ratio; CI = Confidence Interval; NE = Not estimable (0 cases in one of the two groups). * Adjusted for age, gender, hypertension, diabetes, major cardiovascular diseases, chronic obstructive pulmonary diseases, kidney diseases, cancer. ^ψ^ Not estimated for these delayed outcomes, because of the short average follow-up of these subjects—13.0 days for BNT162b2; 17.4 days for mRNA-1273. The average follow-up after a single dose of ChAdOx1 nCoV-19 was 30.4 days. ^A^ Administered at least 14 (BNT162b2 or mRNA-1273) or 21 (ChAdOx1 nCoV-19) days before the end of follow-up (21 May 2021). ^B^ Those who received the second dose were excluded from the analysis; the overall sample was thus composed of 212,371 subjects. ^C^ The comparative analysis of the second dose was restricted to a total of 206,878 subjects—13,414 were excluded as they received the second dose less than 14 days before the end of the follow-up (21 May 2021); 25,364 were excluded because they only received the first dose of vaccine; 181 were excluded because they had a SARS-CoV-2-positive swab after the start of the follow-up of the first dose (14 or 21 days after the first dose, see point C, Table 1) but before the start of the follow-up of the second dose (14 days after 2nd dose administration).

**Table 3 vaccines-09-00628-t003:** Main characteristics and outcomes of the sample, stratified by gender and vaccine status.

Variables	Overall Sample	≥1 Vaccine Dose	Unvaccinated
SARS-CoV-2 positive swab, % (*n*) ^A^			
- Females (*n* = 127,839)	2.76 (3527)	0.13 (53)	3.93 (3474)
- Males (*n* = 117,387)	2.99 (3506)	0.11 (32)	3.97 (3474)
Covid-19 disease, % (*n*) ^A^			
- Females	0.31 (401)	0.03 (12)	0.44 (389)
- Males	0.47 (550)	0.02 (6)	0.62 (544)
Death, % (*n*) ^A^			
- Females	0.09 (109)	0.01 (2)	0.12 (107)
- Males	0.12 (135)	0.00 (1)	0.15 (134)
**Cox model ***	**SARS-CoV-2**	**COVID-19**	**Death**
	HR (95% CI)	HR (95% CI)	HR (95% CI)
≥1 Vaccine dose vs. none ^B^			
- Females (*n* = 39,550)	0.06 (0.04–0.08)	0.06 (0.03–0.11)	0.04 (0.01–0.15)
- Males (*n* = 29,989)	0.04 (0.03–0.06)	0.02 (0.01–0.05)	0.02 (0.00–0.11)
Only one vaccine dose vs. none ^C^			
- Females (*n* = 20,529)	0.16 (0.12–0.22)	0.44 (0.24–0.79)	0.28 (0.04–2.04)
- Males (*n* = 16,155)	0.16 (0.11–0.23)	0.20 (0.09–0.44)	0.27 (0.04–1.94)
Two vaccine doses only vs. none ^D^			
- Females (*n* = 19,021)	0.03 (0.02–0.05)	0.01 (0.00–0.06)	0.04 (0.01–0.27)
- Males (*n* = 13,834)	0.01 (0.01–0.03)	0.00 (NE)	0.00 (NE)

HR = Hazard Ratio; CI = Confidence Interval; NE = Not estimable (0 cases in one of the two groups). * Adjusted for age, hypertension, diabetes, major cardiovascular diseases, chronic obstructive pulmonary diseases, kidney diseases, cancer. ^A^ After 14 (BNT162b2 or mRNA-1273) or 21 (ChAdOx1 nCoV-19) days from the administration of the first dose for the vaccinated subjects, and after 16 January 2021 for the unvaccinated subjects. ^B^ Administered at least 14 (BNT162b2 or mRNA-1273) or 21 (ChAdOx1 nCoV-19) days before the end of follow-up (21 May 2021). ^C^ Those who received the second dose were excluded from the analysis; the overall sample was thus composed of 212,371 subjects. ^D^ The comparative analysis of the second dose was restricted to a total of 206,878 subjects—13,414 were excluded as they received the second dose less than 14 days before the end of the follow-up (21 May 2021); 25,364 were excluded because they only received the first dose of vaccine; 181 were excluded because they had a SARS-CoV-2-positive swab after the start of the follow-up of the first dose (14 or 21 days after the first dose, see point C, Table 1) but before the start of the follow-up of the second dose (14 days after 2nd dose administration).

**Table 4 vaccines-09-00628-t004:** Comparison between those who received the second dose of vaccine within the recommended number of days, and those who had a delayed immunization.

Variables	Recommended Schedule *	Delayed Schedule *	*p* **
	(*n* = 32,164; 97.9%)	(*n* = 691; 2.1%)	
SARS-CoV-2 positive swab, % (*n*) ^A^	0.05 (17)	0.00 (0)	0.5
Covid-19 disease, % (*n*) ^A^	0.00 (1)	0.00 (0)	0.9
Death, % (*n*) ^A^	0.00 (1)	0.00 (0)	0.9

^A^ After 14 days from the administration of the second dose, until the end of follow-up (21 May 2021). * In the recommended schedule, the delay between dose 1 and dose 2 is 21 days (up to a maximum of 28 days) for BNT162b2 vaccine; 28 days for mRNA-1273 vaccine. All subjects who received a second dose beyond this time lag were included into the “delayed schedule” group. ** Fisher’s exact test.

## Data Availability

The data presented in this study are available upon reasonable request from the corresponding author.

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
