# Peer review of "Interim Estimates of COVID-19 Vaccine Effectiveness in a Mass Vaccination Setting: Data from an Italian Province"

_vaccines, 2021, doi:10.3390/vaccines9060628_

Round 1
Reviewer 1 Report
The authors describe their interim estimates of COVID-19 vaccine effectiveness in a mass vaccination roll-out which is extremely encouraging. The benefits of these vaccines are substantial, and their rapid roll-out is an important achievement for public health. To publish these results will be beneficial for reassuring the general population on efficacy and safety, thereby reducing vaccine hesitancy.
I only have minor suggestion to add:
- Methods: The authors describe a follow-up of vaccinated and unvaccinated subjects. How exactly was the follow-up performed?
- Results: The authors describe their interim analysis of vaccine effectiveness. However, it would be interesting to get an impression of the vaccine roll-out itself in the region as Haas et al. (Lancet 2021) as well as Vasileiou et al. (Lancet 2021) report in their publications regarding vaccination in Israel and Scotland. Therefore, the authors may consider to illustrate this in an additional figure. As they here report their findings from an interim analysis, this may also be postponed to the final report.
- Discussion: Considering Italy’s (among others) decision to space vaccine doses by delaying the boost, the here presented data are important. Yet, the need for single-dose person-time follow-up will be of importance and may be discussed. The investigations rely on observational data, but with large data sets as this one comes bias and its limitations. Therefore, please add a comment on recall or misclassification bias. How may the testing policy have influenced differences of vaccinated vs. unvaccinated? Further: How did non-vaccine effect the results such as hospital admissions (lock-down or other restrictions)?
Author Response
I-1. The Referee wrote "The authors describe their interim estimates of COVID-19 vaccine effectiveness in a mass vaccination roll-out which is extremely encouraging. The benefits of these vaccines are substantial, and their rapid roll-out is an important achievement for public health. To publish these results will be beneficial for reassuring the general population on efficacy and safety, thereby reducing vaccine hesitancy.
I only have minor suggestion to add:
Methods: The authors describe a follow-up of vaccinated and unvaccinated subjects. How exactly was the follow-up performed?".
We agree and thank the Referee for the comments. We accordingly added the following paragraph to the Methods section, which was titled "Public health surveillance".
"The Italian Ministry of Health provided the general guidelines for the National immunization campaign [8]. According to this plan, each Local Health Unit is in charge of administering the vaccines and provide all healthcare services to its residents. The surveillance of vaccine uptake, as well as of SARS-CoV-2 testing and COVID-19 cases are part of the National pandemic response plan, and all the information about vaccines, laboratory tests, demographic, anamnestic and clinical residents' data are routinely entered into LHU official datasets, updated daily, and are sent to the Italian Institute of Health [20].
Vaccination, SARS-CoV-2 testing, and primary or hospital care are free of charge. Regardless of vaccination status, tests are mandatory for all individuals with suggestive symptoms such as fever or acute respiratory illness, for those that have been in contact with infected persons, and for all individuals returning from travel abroad [8].
Please acknowledge that the following reference was added:
[20] Riccardo F, Andrianou X, Bella A et al. Italian National Institute of Health. COVID-19 integrated surveillance system. Web Page. Last update: May 29, 2021. https://www.epicentro.iss.it/coronavirus/sars-cov-2-sorveglianza
I-2. The Referee wrote "Results: The authors describe their interim analysis of vaccine effectiveness. However, it would be interesting to get an impression of the vaccine roll-out itself in the region as Haas et al. (Lancet 2021) as well as Vasileiou et al. (Lancet 2021) report in their publications regarding vaccination in Israel and Scotland. Therefore, the authors may consider to illustrate this in an additional figure. As they here report their findings from an interim analysis, this may also be postponed to the final report"
We thank the Reviewer for the suggestion. Please acknowledge that, as mentioned into the previous answer, we briefly described the organization of the vaccination campaign into a new, dedicated paragraph, which was added to the Methods section. As regards vaccine roll-out, given that this study is based upon preliminary data, and that the vaccination campaign is still ongoing, with data substantially changing daily, we entirely agree that a detailed figure, displaying vaccine uptake data, could be better suited for the final report. In this regard, we believed that it would have been important to update the data as much as we could, and we thus collected the information on those vaccinated, tested and hospitalized during the manuscript submission process. The end of follow-up was thus extended from April 26 to May 21, and besides the longer follow-up, this permitted the inclusion of an additional 32,072 vaccinated subjects (from the previous 37,467 to the current 69,539). All analyses were thus remade, and although the results are similar, all numbers have been updated throughout the manuscript and tables.
Please also acknowledge that during this time we had the chance to add the information, for all vaccinated and unvaccinated residents, on some relevant comorbidities including hypertension, diabetes, major cardiovascular and cerebrovascular events, chronic obstructive pulmonary diseases, kidney diseases and cancer. We thus adjusted all multivariate analyses for these comorbidities, in order to reduce the potential bias in the evaluation of the association between vaccination and severe or lethal COVID-19.
Finally, to be conservative, we excluded from the analyses the subjects that are registered in the healthcare system of the Local Health Unit of the Province of Pescara, but they do not reside in this Province. In the Italian healthcare system, especially during the current period of movement restrictions, these persons might have received a SARS-CoV-2 test in their Province of residence instead of the Province where they opted to be registered for the healthcare services, and we were unfortunately unable to retrieve such information (while we had all the data of the residents). The overall sample thus decreased, and currently consists of 69,539 vaccinated and 175,687 unvaccinated adult residents.
We understand that this last point might be quite confusing, but please acknowledge that in Italy we have two different anagraphic datasets: one of the healthcare system, which can be changed by each person in every moment, and the one of the census, which is linked to the address of residence and cannot be changed unless the address is modified. A person is free to opt to be registered in one Local Health Unit regardless of his/her residence, and even of his/her domicile or work address. As a result, in each Local Health Unit there is a non trivial proportion of subjects who are registered as "healthcare residents", but are not "census residents". Being registered into a Local Health Unit healthcare anagraphic only implies that the Local Health Unit must provide the care you eventually need, but all the registered citizens are still free to chose and be treated free-of-charge in any other Local Health Unit. This is why it may be relatively common, especially during a period in which movement restrictions are applied, that persons who are registered into a Local Health Unit, but reside in the territory of another Local Health Unit, opt to receive some services in the Local Health Unit of residence instead of the Unit in which they are registered.
I-3. The Referee wrote "Discussion: Considering Italy’s (among others) decision to space vaccine doses by delaying the boost, the here presented data are important. Yet, the need for single-dose person-time follow-up will be of importance and may be discussed. The investigations rely on observational data, but with large data sets as this one comes bias and its limitations. Therefore, please add a comment on recall or misclassification bias. How may the testing policy have influenced differences of vaccinated vs. unvaccinated? Further: How did non-vaccine affect the results such as hospital admissions (lock-down or other restrictions)?"
We entirely agree that studies relying on observational data are prone to recall or misclassification bias. Similarly, if the testing policies are influenced by vaccination, this may severely bias this results. Finally, lockdown or other restrictions may have affected the results, especially on hospital admissions. We accordingly added the following paragraph to the Discussion: "Also, although our analyses were based upon vaccination and clinical data extracted from routinely collected electronic health records, yet observational studies are prone to recall or misclassification bias [24], and biases could be introduced as well if testing policies were affected by vaccination status, or restrictive measures were applied differently to vaccinated and unvaccinated subjects. Although misclassification bias is unlikely for COVID-19 cases and deaths, it is possible that some SARS-CoV-2-infected individuals, who reported being asymptomatic at the time of interview, might have instead been presymptomatic. However, as reported by Haas and Colleagues, this type of misclassification was probably uncommon and would be unlikely to substantially influence the vaccine effectiveness estimate against asymptomatic infection [23]. With regard to testing and restriction policies, in Italy, in contrast to other countries [23], vaccinated and unvaccinated subjects were subject to the same testing requirements and restriction policies, thus being unlikely that the estimates are substantially affected by test frequency or lockdown measures [42]".
Please acknowledge that the following references were added:
[23] Haas EJ, Angulo FJ, McLaughlin JM et al. Impact and effectiveness of mRNA BNT162b2 vaccine against SARS-CoV-2 infections and COVID-19 cases, hospitalisations, and deaths following a nationwide vaccination campaign in Israel: an observational study using national surveillance data. Lancet 2021;397:1819-29.
[24] Vasileiou E, Simpson CR, Shi T et al. Interim findings from first-dose mass COVID-19 vaccination roll-out and COVID-19 hospital admissions in Scotland: a national prospective cohort study. Lancet 2021; 397:1646-57.
[42] Italian National Institute of Health. COVID-19 Vaccines FAQ. Available online: https://www.iss.it/covid19-faq/ (accessed on June 2, 2021).
Reviewer 2 Report
In this interim study, Manzoil et.al, reported the real-world data of three COVID-19 vaccines from an Italian province. The effectiveness of two doses of vaccination, one dose vaccination and unvaccinated, as well as effectiveness between genders, were compared. The outcomes, including RT-PCT detection of SARS2 infection, COVID-19 disease, and COVID-19-related death, were analyzed. The study is well designed with results clearly presented and conclusion-supported. The manuscript is well written and the findings support the current immunization policy. The reviewer has only minor comments:
1) Line 29, Novel coronavirus disease. It would be better to use " Coronavirus disease-2019"
2) Line 55, please define RCTs in its first appearance
3) Line 107-108 and Table 1, 5878 (2.15%) among the unvaccinated. The percentage 2.15% is not consistent with the value 2.47 in Table 1.
4) Line 109, COVID-19 cases were 776;773 should be 773 ?
5) Line 123: The observation that "there was no risk reduction among those receiving a single dose of BNT162b2." This is interesting. Could the authors add more comments on the information of single-dose for the mRNA-1273 comparatively with the BNT162b2, as the contrast between these two was so striking?
6) Line 263, "By now, several laboratory assessments... in vitro activity.." I would suggest being more specific by using "in vitro antibody neutralizing activity".
Author Response
II-1. The Referee wrote "In this interim study, Manzoli et. al reported the real-world data of three COVID-19 vaccines from an Italian province. The effectiveness of two doses of vaccination, one dose vaccination and unvaccinated, as well as effectiveness between genders, were compared. The outcomes, including RT-PCR detection of SARS2 infection, COVID-19 disease, and COVID-19-related death, were analyzed. The study is well designed with results clearly presented and conclusion-supported. The manuscript is well written and the findings support the current immunization policy. The reviewer has only minor comments:
1) Line 29, Novel coronavirus disease. It would be better to use "Coronavirus disease-2019"
2) Line 55, please define RCTs in its first appearance
3) Line 107-108 and Table 1, 5878 (2.15%) among the unvaccinated. The percentage 2.15% is not consistent with the value 2.47 in Table 1.
4) Line 109, COVID-19 cases were 776;773 should be 773?
5) Line 263, "By now, several laboratory assessments... in vitro activity.." I would suggest being more specific by using "in vitro antibody neutralizing activity".
We agree and thank the Referee for the comments and suggestions. Please acknowledge that we amended the text accordingly, and corrected the errors, of which we apologize. In specific:
1. Line 29: "Novel coronavirus disease" was replaced with "Coronavirus disease-2019".
2. The acronym RCTs was added after its first mention in the text (line 33).
3. Line 141 (previously lines 107-108) and Table 1: the 2.15% value reported in the text has been replaced with the new correct value "4.00%". Please acknowledge that, in this rapidly evolving scenario, with huge amount of data being accumulated daily, we believed that it would have been important to update the data as much as we could, and we thus collected the information on those vaccinated, tested and hospitalized during the manuscript submission process. The end of follow-up was thus extended from April 26 to May 21, and besides the longer follow-up, this permitted the inclusion of an additional 32,072 vaccinated subjects (from the previous 37,467 to the current 69,539). All analyses were thus remade, and although the results are similar, all numbers have been updated throughout the manuscript and tables.
Please also acknowledge that during this time we had the chance to add the information, for all vaccinated and unvaccinated residents, on some relevant comorbidities including hypertension, diabetes, major cardiovascular and cerebrovascular events, chronic obstructive pulmonary diseases, kidney diseases and cancer. We thus adjusted all multivariate analyses for these comorbidities, in order to reduce the potential bias in the evaluation of the association between vaccination and severe or lethal COVID-19.
Finally, to be conservative, we excluded from the analyses the subjects that are registered in the healthcare system of the Local Health Unit of the Province of Pescara, but they do not reside in this Province. In the Italian healthcare system, especially during the current period of movement restrictions, these persons might have received a SARS-CoV-2 test in their Province of residence instead of the Province where they opted to be registered for the healthcare services, and we were unfortunately unable to retrieve such information (while we had all the data of the residents). The overall sample thus decreased, and currently consists of 69,539 vaccinated and 175,687 unvaccinated adult residents.
We understand that this last point might be quite confusing, but please acknowledge that in Italy we have two different anagraphic datasets: one of the healthcare system, which can be changed by each person in every moment, and the one of the census, which is linked to the address of residence and cannot be changed unless the address is modified. A person is free to opt to be registered in one Local Health Unit regardless of his/her residence, and even of his/her domicile or work address. As a result, in each Local Health Unit there is a non trivial proportion of subjects who are registered as "healthcare residents", but are not "census residents". Being registered into a Local Health Unit healthcare anagraphic only implies that the Local Health Unit must provide the care you eventually need, but all the registered citizens are still free to chose and be treated free-of-charge in any other Local Health Unit. This is why it may be relatively common, especially during a period in which movement restrictions are applied, that persons who are registered into a Local Health Unit, but reside in the territory of another Local Health Unit, opt to receive some services in the Local Health Unit of residence instead of the Unit in which they are registered.
4. Lines 142-143 (previously lines 109-110): please acknowledge that, to increase the clarity of the sentence, the previous lines "COVID-19 cases were 776; 773 of whom in the unvaccinated group (0.33%), and 3 among vaccinated subjects" was rephrased as follows: "COVID-19 cases were 951; of these, 933 were registered in the unvaccinated group (0.53%), and 18 among vaccinated subjects (0.03%; p<0.001)". As above, all numbers have been updated and rechecked, we apologize for the oversights.
5. Line 316 (previously line 263): "in-vitro activity" has been replaced with "in-vitro antibody neutralizing activity".
II-2. The Referee also wrote "Line 123: The observation that "there was no risk reduction among those receiving a single dose of BNT162b2". This is interesting. Could the authors add more comments on the information of single-dose for the mRNA-1273 comparatively with the BNT162b2, as the contrast between these two was so striking?".
We thank the Referee for the suggestion and we entirely agree that the result was striking. This lead us to verify all data, and we realized that the average duration of the follow-up of the subjects who received only one dose of m-RNA vaccines was too short (13.0 days for BNT162b2; 17.4 days for mRNA-1273) to permit a meaningful evaluation of the effectiveness of these vaccines against outcomes that may require 3-20 days to occur after the contagion. As the mean follow-up of those receiving a single dose of ChAdOx1 nCoV-19 was 30.4 days, we kept only the analyses of the effectiveness after a single dose of ChAdOx1 nCoV-19. Please acknowledge that we accordingly updated the Results, Table 2, and Discussion, as follows: "While the average follow-up of those receiving a single dose of ChAdOx1 nCoV-19 was 30.4 days, the mean follow-up duration of the subjects who received only one dose of m-RNA vaccines was too short (13.0 days for BNT162b2; 17.4 days for mRNA-1273) to permit a meaningful evaluation of the effectiveness of these vaccines against outcomes, such as COVID-19 or death, that may require 3-20 days to occur after the contagion [21].
Please acknowledge that the following reference was added:
[21] Flacco ME, Acuti Martellucci C, Bravi F, et al. Severe Acute Respiratory Syndrome Coronavirus 2 Lethality Did not Change Over Time in Two Italian Provinces. Open Forum Infect Dis 2020, 7, ofaa556, doi:10.1093/ofid/ofaa556ofaa556.